# Work ability during the COVID-19 pandemic: A cross-sectional study in a low-income urban setting in Brazil

Ana Paula Cândido Oliveira[1,2], Daniela Alencar Vieira[1,2], Cristiane Wanderley Cardoso[1,2], Tereza Magalhães[1,3], Rosangela Oliveira Anjos[4], Eduardo José Farias Borges Reis[1], Kionna Oliveira Bernardes Santos[1], Guilherme Sousa Ribeiro [1,4]*

1 Faculdade de Medicina, Universidade Federal da Bahia, Salvador, Bahia, Brazil, 2 Secretaria Municipal de Saúde de Salvador, Salvador, Bahia, Brazil, 3 Texas A&M University, College Station, Texas, United States of America, 4 Instituto Gonçalo Moniz, Fundação Oswaldo Cruz, Salvador, Bahia, Brazil

* guilherme.ribeiro@fiocruz.br

## Abstract

Work ability is a subjective concept that reflects the balance between an individual's perception of the physical, mental, and social demands of work and their competence and resources to meet those demands. The COVID-19 crisis significantly impacted health, work, and socioeconomic conditions worldwide. However, few studies have examined work ability in disadvantaged urban communities during this period. To analyze factors associated with work ability within the context of social vulnerability during the COVID-19 pandemic, we conducted a cross-sectional study in a low-income neighborhood in Salvador, Brazil, between February and June 2022. Socio-demographic, health, and labor data were collected, and work ability was assessed using the Work Ability Index (WAI), a widely used tool for evaluating work ability. Multivariable analyses based on a hierarchical model were run to investigate factors associated with low WAI scores. The study included 292 workers aged ≥16 years (59.6% women; median age 41 years). Most workers (84.6%) were classified as having adequate work ability based on their WAI scores. Multivariable analyses found that inadequate work ability was more frequent among women (prevalence ratio [PR]: 1.89, 95% confidence interval [CI]: 1.02-3.48), individuals who self-rated their health as moderate/good (PR: 5.91; 95% CI: 1.45-24.05) or poor/very poor (PR: 21.62; 95% CI: 5.14-90.91) compared to those with excellent/very good health, and those reporting diabetes (PR: 2.1; 95% CI: 1.13-3.9). Working >40 hours per week (PR: 0.47; 95% CI: 0.28-0.96) was negatively associated with inadequate work ability, suggesting that individuals with adequate work ability may be selected for longer working hours. A history of COVID-19 was not associated with inadequate work ability. These findings suggest that targeted interventions to improve work ability in low-income communities should prioritize women and workers with chronic health conditions, such as diabetes.

**Data availability statement:** All relevant data are within the paper and its Supporting Information files.

**Funding:** This study was supported by the Secretaria Municipal de Saúde de Salvador (Salvador Municipal Health Department), Brazil; the Fundação Oswaldo Cruz, Ministério da Saúde (Oswaldo Cruz Foundation, Ministry of Health) Brazil, and the Universidade Federal da Bahia (Federal University of Bahia), Brazil. The Conselho Nacional de Desenvolvimento Científico e Tecnológico (National Council for Scientific and Technological Development, CNPq) provided research scholarship to GSR (grant 311365/2021-3). The funders had no role in study design, data collection and analysis, decision to publish, or preparation of the manuscript.

**Competing interests:** The authors have declared that no competing interests exist.

## Introduction

The COVID-19 pandemic had both direct and indirect consequences worldwide. Beyond its health impacts, the pandemic and the measures taken to contain its spread exacerbated existing social vulnerabilities, such as poverty, unequal access to education, and precarious working conditions [1–3]. In the workforce, essential professionals, including those in health care, transportation, media, and food services, faced overwhelming demands and often increased working hours [2–5]. The mental strain from fears of contagion, heightened workloads, and staff shortages further contributed to an exhausting work environment, with significant consequences for mental health [2–7]. Sectors with a lower risk of direct exposure to contamination also had to adapt to the new realities, facing challenges such as the transition to remote work (often without adequate regulations), reduced paid working hours, employment contract suspensions, and widespread layoffs [3,6,8–11].

Across the world, several support and intervention policies were implemented to mitigate the impact of the labor market crisis caused by the pandemic [1,10]. In Brazil, the Emergency Employment and Income Maintenance Program played a crucial role in temporarily suspending employment contracts and reducing working hours with salary compensation [8]. These measures aimed to preserve jobs and income for formal workers while helping companies navigate the economic crisis without resorting to mass layoffs [12]. For the most socially vulnerable groups – comprising informal workers, individual microentrepreneurs, the self-employed, unemployed individuals, and families benefiting from Bolsa Familia, a federal income transfer program – an Emergency Aid subsidy was introduced to provide a minimum income [13]. Despite the importance of these social policies, their effectiveness varied widely, and they were unable to reach all those affected [14,15]. As a result, pre-existing social vulnerabilities worsened, leading to increased poverty, unemployment, informal work, gender wage gap, and reduced access to education [1,5,10,16–18].

In this context, it was expected that the economic uncertainties and professional instability imposed by the COVID-19 pandemic would affect work ability globally, especially among the most vulnerable workers. Work ability is a multifaceted attribute that encompasses more than just technical or physical skills. It also considers an individual's adaptability to work demands and their ability to maintain satisfactory performance despite changes in health conditions or the work environment [19]. It is defined as a reflection of a worker's immediate and future well-being, closely linked to the physical, mental, and social demands of work activity [20–22]. Thus, work ability is a central element of both well-being and working life.

However, to our knowledge, no studies have examined work ability in low-income urban settings during the COVID-19 pandemic. In this study, we assessed the work ability of residents in an informal urban neighborhood in Brazil shortly after the peak of transmission of the Omicron BA.1 and BA.2 SARS-CoV-2 subvariants [23] and investigated factors associated with inadequate work ability.

## Method

### Study design and setting

This cross-sectional study was conducted from February 23 to June 22, 2022, in the Alto das Pombas neighborhood, an informal urban settlement in Salvador, Brazil. Alto das Pombas has a low-income population: 53% of residents registered in the Family Health Strategy (FHS) have 8 or fewer years of education, and 52% of the families earn up to the minimum Brazilian wage per month (US$ 254 as of June 2022) (unpublished data from the FHS, Secretary of Health of Salvador). The community is served by the governmental FHS, which provides free preventive and primary health care centered on individuals and families in its territory. Each of the two FHS teams serving the community consists of one physician, one nurse, two nursing technicians, and six community health agents [24].

Participants were enrolled based on the following criteria: age ≥ 16 years, residency in the community (as defined by registration in the local FHS), and the ability to understand and respond to the survey questions. Although the Brazilian Apprenticeship Law [25] permits individuals as young as 14 years to work, we set the inclusion age limit at 16 years to avoid selecting young adolescents, whose likelihood of working was minimal. The study included both working and non-working individuals, enabling an examination of the employment context during the pandemic.

To obtain a representative sample of the neighborhood population, we randomly selected 630 individuals from the 5,478 residents aged 16 years or older registered in the community's FHS, distributed across 11 microareas. Sampling was stratified by microarea and weighted according to the proportion of the population in each microarea. Random selection was performed at the household level using Excel (Office 365®), based on the FHS household registry. Considering an average household size of approximately three individuals, we randomly selected a number of households equivalent to about one-third of the weighted estimated number of required individuals per microarea. All eligible individuals aged 16 or older in the selected households were invited to participate.

This study was part of a broader research project investigating health and living conditions during the COVID-19 pandemic. The initial sample size of 630 participants was calculated to provide approximately 4% precision for estimating events with a 50% prevalence at a 95% confidence level. However, due to logistical challenges and restrictions imposed by the pandemic, the final sample comprised 504 participants from 273 households, including 292 workers residing in 205 households.

### Data collection

The study team conducted home-visit interviews to collect sociodemographic, health, and occupational data using a mobile-based structured questionnaire on the Research Electronic Data Capture (REDCap®) software. If a participant could not be interviewed at home, the interview was conducted at the local FHS unit. Sociodemographic data collected included sex, age, marital status, education level, self-declaration of race/skin color, and income. Family-level data, such as enrollment in the Bolsa Familia program (a conditional cash transfer program provided by the federal government to families living in poverty) and in any emergency aid programs established in 2020 and 2021 due to the COVID-19 pandemic, were also obtained.

Self-reported health data included weight, height, general health status (based on a five-item Likert scale to the question "*In general, how would you say your health is*?"), prior diagnoses of hypertension and diabetes, smoking history, and history of COVID-19. Occupational data included working status, type and number of job/occupations, weekly working hours, work as a young apprentice, working from home, retirement history, changes in income during the pandemic, level of work-related exposure to SARS-CoV-2 infection [26], loss or change of work since the beginning of the pandemic, and work ability. Work ability was assessed using the Brazilian Portuguese-translated and adapted version of the Work Ability Index (WAI) [22], which consists of ten questions across seven dimensions: 1) current work ability relative to the lifetime best, 2) work ability relative to job demands, 3) number of current diseases, 4) estimated work impairment due to diseases, 5) sick leave in the past year, 6) own prognosis of work ability in two years, and 7) mental resources [20]. The WAI

was developed by Finnish researchers to evaluate how long individuals should continue working [27,28] and has been widely used to assess work ability [20,29–33].

## Data analysis

Descriptive analyses were performed to characterize the study participants by work status (with or without occupation), using absolute and relative frequencies, as well as measures of central tendency and dispersion. Age was categorized into two groups (<45 and ≥45 years), based on the World Health Organization's definition of "aging worker", which considers individuals ≥45 years as potentially having reduced functional working capacities [19,34]. Individual monthly income was categorized into two groups: up to BRL 1,000 and more than BRL 1,000. This cutoff (approximately USD 210) was chosen to create two groups of roughly equal size, using a rounded and meaningful BRL value. Work-related exposure risk to SARS-CoV-2 was classified into four levels: very high (health professionals performing aerosol-generating procedures), high (health professionals exposed to COVID-19 patients without performing such procedures), medium (professionals with close contact with potentially infected individuals), and low (professionals without significant contact with suspected or confirmed cases) [26].

The WAI score was calculated as the sum of points from the instrument's seven dimensions and initially categorized into four levels: low (7–27), moderate (28–36), good (37–43), and excellent (44–49) [21]. The scores were then recategorized into two levels: inadequate work ability (<37, combining the low and moderate categories), and adequate work ability (≥37, combining the good and excellent categories). This binary classification is supported by the WAI's conceptual framework, which recommends that individuals with inadequate work ability (<37) receive targeted measures to restore or improve their work capacity, whereas those with adequate work ability (≥37) should focus on maintaining and supporting it. This dichotomization has also been applied in previous research [35–37].

To investigate factors associated with inadequate work ability, while accounting for the complex interrelationships between its determinants, we used a theoretical framework based on hierarchical relationships between the potentially explanatory variables and the dependent variable (inadequate work ability) [38]. This framework was developed using various multidimensional and integrated models of work ability [21]. It included three levels of related factors, considering both the hierarchical interrelationships among them and their connection to the dependent variable (Fig 1). In addition, demographic characteristics such as sex and age were included in the model for adjustment.

The most distal level included personal sociodemographic and economic social vulnerability indicators (e.g., race/skin color, education level, income, support from government income transfer programs – such as Bolsa Familia and Pandemic Emergency Aid – and marital status). These variables can potentially exert a direct influence on the other two levels of factors: the intermediate level, which includes occupational characteristics and work patterns (e.g., homeworking, number of working hours, employment status [formal vs informal], number of jobs, being a young apprentice, being retired and working, changes in employment and income during the pandemic); and the proximal level, which addresses health aspects (self-reported diabetes, high blood pressure, smoking, body mass index and prior COVID-19). Health aspects have been shown to significantly impact work ability [30]. Furthermore, specific questions on health status are included in the work ability construct (WAI), justifying this health block as the most proximal.

According to the theoretical framework model, bivariate and multivariable Poisson regression analyses with robust variance were used to identify factors associated with inadequate work ability. This method was chosen because it allows direct estimation of prevalence ratios (PRs), which are more appropriate and interpretable than odds ratios in cross-sectional studies with moderately or highly prevalent outcomes. First, bivariate analyses were performed to estimate PR and 95% confidence intervals (CI) and to select variables related to work ability at a significance level of 0.20 (P < 0.20) within each hierarchical level. Next, a series of multivariable analyses with backward elimination and adjustment for age and sex were run for each of the three blocks of variables (from the most distal to the most proximal level), including the within-block variables identified in the bivariate analyses. The backward elimination strategy was adopted to remove within-block variables that do not contribute meaningfully to explaining the outcome, thereby reducing model complexity

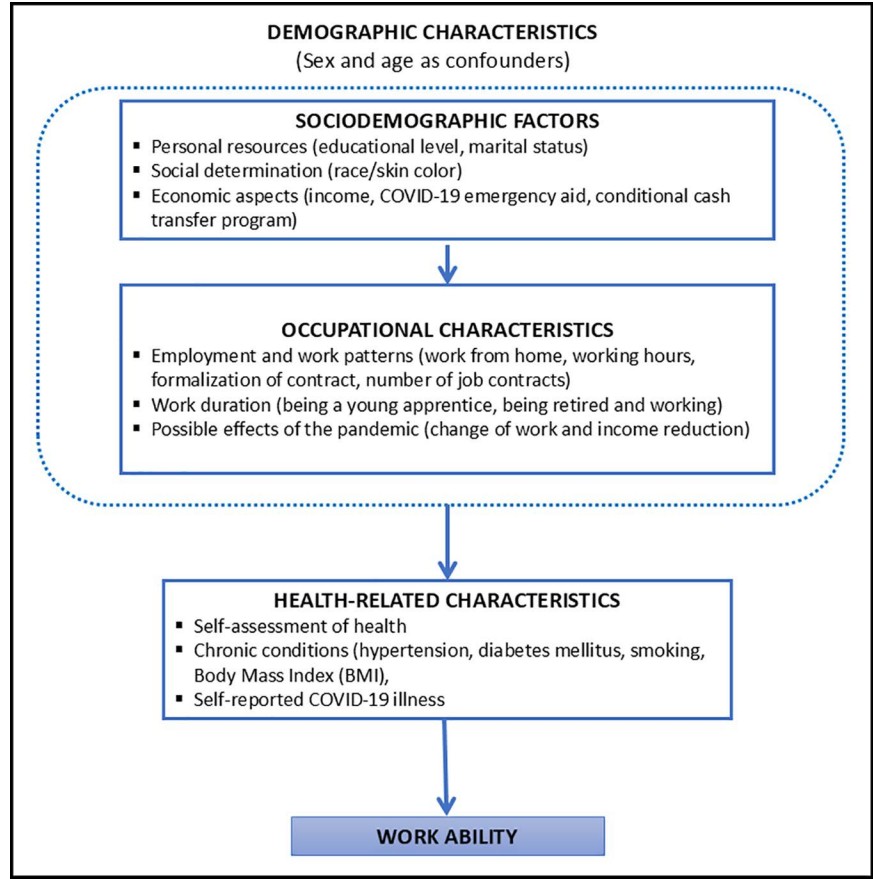

**Fig 1. Theoretical framework for the hierarchical interrelationship of social, occupational, and health characteristics with work ability.**

and minimizing redundancy due to correlations among conceptually related variables within the same domain (e.g., socio-economic or health conditions).

The first multivariable model included only the selected variables from the sociodemographic and economic block (distal level). Variables showing an association with work ability at a $P < 0.05$ were retained and included in the analyses of the intermediate and proximal levels, regardless of the P values obtained in these subsequent analyses. Similarly, selected variables from the occupational block (intermediate level) that showed an association with $P < 0.05$ were retained and included in the health block (proximal level) analysis, regardless of the P values at this level. Variables associated with work ability at a $P < 0.05$ in any of the blocks were defined as independently associated with work ability. Adjusted PR and 95% CI were calculated for the variables retained in each of the three blocks. The analyses were performed using IBM SPSS software version 20.0.0. The analyzed database and corresponding codebook are available in the supporting information files (S1 Database and S1 File).

### Ethical aspects

All participants aged ≥18 years, as well as the legal guardians of a participant <18 years, signed an informed consent form. Participants <18 years old also signed an informed assent form. The study was approved by the Research Ethics Committee of the School of Medicine of Bahia, Federal University of Bahia (CAAE 55012022.5.0000.5577).

## Results

### Characteristics of the study participants

Of the 504 participants, 344 (68.3%) were female, and 470 (93.3%) self-reported as black or mixed race/skin color. The mean age was 45 years (standard deviation, SD = 17). Overall, education levels were low: 259 (51.4%) participants had not studied beyond 8 years of schooling. The majority (292; 57.9%) of participants were working, though a substantial portion (212; 42.1%) were not (Table 1). Compared to participants who were not working, those who were working were younger, included a higher proportion of men, were more likely to have a partner, and had more years of education (Table 1). A history of COVID-19 was more commonly reported among those who worked, with no participants reporting hospital admission due to the illness. In contrast, diabetes and hypertension were more prevalent among those who did not work. Among the 81 workers with at least one of these two comorbidities, more than half (62; 76.5%) were aged 45 or older. Notably, 11 (61.1%) of the 18 workers with diabetes also had hypertension.

The reasons for not being in paid work varied among the 212 participants who were not working. Of these, 63 (29.7%) were actively seeking employment, and 49 (23.1%) reported being retired. Other reasons included health-related issues, including pregnancy (26; 12.2%), not wanting to work (22; 10.4%), dedication to studies (19; 9.1%), and household chores or caregiving for family members (16; 7.5%). Dedication to studies was primarily mentioned by men (17 of the 19), whereas household chores and caregiving were exclusively cited by women (16 of the 16). Social distancing due to the pandemic was reported as a reason by 5 (2.4%) participants. Additionally, 30 (14.2%) participants reported other reasons. The total exceeds 212 because some participants provided more than one reason for not working.

Additionally, 76 (26.0%) of the 292 participants who were currently working, reported changing occupations since the start of the pandemic. Among these 292 workers, 105 (36.0%) had formal employment (with a work contract and national social security registration), while 187 (64.0%) were in informal jobs (without a contract or social security registration). Most workers reported having only one work activity (236, 80.8%) and working up to 40 hours per week (193; 66.1%). Of the 276 workers who responded on their income, 119 (43.1%) earned less than BRL 1,000 (USD ~ 210) per month). The average monthly income was BRL 1,272.35, SD: 931.18, range: 100.00–6,000.00 (USD 267.34, SD: 194.64; range: 20.93 − 1,256.15).

The predominant employment sector was cleaning (59, 20.2%), followed by commerce (57, 19.5%), health (28, 9.6%), beauty and clothing (20, 6.8%), administrative services (16, 5.5%), delivery and transportation (15, 5.1%), security (15, 5.1%), construction (14, 4.8%), food services (11, 3.8%), education (8, 2.7%), and others (62, 21.2%); the sum of these numbers exceeds 292 because some participants had multiple work activities. Additionally, 41 (14.0%) participants reported working from home, primarily in food production, sales, beauty, education, and crafts; all of whom were women.

### Work ability level

The mean WAI for participants who were working was 41.6 (SD: 5.6), with a median of 43. Adequate work ability (WAI ranging from 37 to 49) was observed in 247 (84.6%) workers, while inadequate work ability (WAI ranging from 7 to 36) was observed in 45 (15.4%) workers. Among those with adequate work ability, 119 (48.2%) had excellent and 128 (51.8%) had good work ability. Among those with inadequate work ability, 37 (82.2%) had moderate and 8 (17.8%) had low work ability.

### Factors associated with inadequate work ability

Inadequate work ability was more common among women, individuals with a monthly income up to USD 210, recipients of the COVID-19 pandemic emergency aid, those working up to 40 hours per week, formally employed workers, individuals with low self-rated health, and those with a prior diagnosis of hypertension or diabetes mellitus (DM) ($P < 0.05$ for all variables; Table 2, in bold). Other variables not significantly associated with inadequate work ability, but included in the

**Table 1. Sociodemographic characteristics of the study population, according to occupational status, Salvador, Brazil, 2022.**

| Characteristics | Total | Working | Not working |
|---|---|---|---|
| | (N: 504) | (N: 292) | (N: 212) |
| | Number (%) | | |
| **Sex** | | | |
| Female | 344 (68.3) | 174 (59.6) | 170 (80.2) |
| Male | 160 (31.7) | 118 (40.4) | 42 (19.8) |
| **Age group** | | | |
| <45 | 257 (51.0) | 167 (57.2) | 90 (42.5) |
| ≥45 | 247 (49.0) | 125 (42.8) | 122 (57.5) |
| **Race/Skin color** | | | |
| Black | 282 (56.3) | 172 (59.3) | 110 (52.1) |
| White, mixed, other | 219 (43.7) | 118 (40.7) | 101 (47.9) |
| **Marital status** | | | |
| Single | 235 (46.6) | 140 (47.9) | 95 (44.8) |
| Married or living with a partner | 216 (42.9) | 137 (46.9) | 79 (37.3) |
| Separated or widowed | 53 (10.5) | 15 (5.1) | 38 (17.9) |
| **Education** | | | |
| Up to 8 years of study | 110 (23.3) | 48 (17.1) | 62 (32.3) |
| 9 to 11 years of study | 119 (23.6) | 58 (20.6) | 61 (31.8) |
| 12 to 15 years of study | 220 (43.7) | 156 (55.5) | 64 (33.3) |
| 16 years or more of study | 24 (4.8) | 19 (6.8) | 5 (2.6) |
| **Body mass index** | | | |
| Underweight | 23 (4.7) | 9 (3.2) | 14 (6.7) |
| Normal weight | 155 (31.5) | 89 (31.3) | 66 (31.7) |
| Overweight | 179 (36.4) | 110 (38.7) | 69 (33.2) |
| Obesity | 135 (27.4) | 76 (26.8) | 59 (28.4) |
| **Self-reported clinical conditions** | | | |
| Smoker | 37 (6.8) | 20 (6.8) | 17 (8.0) |
| Hypertension | 160 (29.5) | 74 (25.3) | 86 (40.6) |
| Diabetes | 51 (9.4) | 18 (6.2) | 33 (15.6) |
| **Self-reported COVID-19 and vaccination history** | | | |
| COVID-19 illness | 166 (32.9) | 104 (35.6) | 62 (29.2) |
| COVID-19 vaccination (one or more doses) | 497 (98.6) | 290 (99.3) | 207 (97.6) |

multivariable model, were age, enrollment in the Bolsa Familia program and self-reported prior COVID-19 illness (P < 0.20; Table 2, in bold).

The first multivariable model using the sociodemographic data (model A, Table 3), showed a higher prevalence of inadequate work ability among women compared to men (PR: 1.89, 95% CI: 1.02–3.48; P = 0.04) and among workers aged ≥45 years compared to those <45 years (PR: 1.69; 95% CI: 0.99–2.89; P = 0.06). The second multivariable model, which added occupational variables to the variables selected in model A (model B, Table 3), found that those working more than 40 hours per week were less likely to have inadequate work ability than those working up to 40 hours per week (RP: 0.47; 95% CI: 0.28–0.96; P = 0.04). The third multivariable model (model C, Table 3), which incorporated health-related variables along with those selected in model B, found higher prevalences of inadequate work ability among those who self-rated their health as good/moderate (PR: 5.91; 95% CI 1.45–24.05; P = 0.01) or bad/very bad (RP: 21.62; 95% CI: 5.14–90.91; P > 0.00), compared to those who self-rated their health as excellent/very good, and among those with diabetes (PR: 2.10;

**Table 2. Frequency of inadequate work ability, according to sociodemographic, occupational, and health-related characteristics, Salvador, Brazil, 2022.**

| Characteristics | Categories | Number of participants | Number (%) with inadequate work ability | Crude PR (95% CI) [1] | P value [1] |
|---|---|---|---|---|---|
| **Sociodemographic** | | | | | |
| Sex | Female | 174 | 33 (19.0) | 1.87 (1.01 - 3.46) | **0.048** |
| | Male | 118 | 12 (10.2) | 1 | |
| Age group | ≥45 years | 125 | 25 (20.0) | 1.27 (0.70 - 2.31) | **0.063** |
| | <45 years | 167 | 20 (12.0) | 1 | |
| Race/ ethnicity | Black | 172 | 25 (14.5) | 0.90 (0.52–1.56) | 0.737 |
| | White, mixed, other | 118 | 19 (16.1) | 1 | |
| Marital status | Not married or living with a partner | 155 | 23 (14.8) | 0.92 (0.54 −1.58) | 0.773 |
| | Married or living with a partner | 137 | 22 (16.1) | 1 | |
| Schooling | Up to 8 years of study | 59 | 12 (20.3) | 1.93 (0.47–7.87) | 0.334 |
| | 9 to 11 years of study | 58 | 5 (8.6) | 0.82 (0.17–3.88) | |
| | 12 to 15 years of study | 156 | 26 (16.7) | 1.58 (0.48–6.15) | |
| | 16 years or more of study | 19 | 2 (10.5) | 1 | |
| Monthly income | Up to USD 210 | 119 | 24 (20.2) | 1.76 (1.00–3.09) | **0.049** |
| | More than USD 210 | 157 | 18 (11.5) | 1 | |
| Bolsa familia | Yes | 32 | 8 (25.0) | 1.76 (0.90 - 3.43) | **0.099** |
| | No | 260 | 37 (14.2) | 1 | |
| COVID-19 emergency aid, 2020 | Yes | 150 | 25 (16.7) | 1.16 (0.67–1.99) | 0.594 |
| | No | 139 | 20 (14.4) | 1 | |
| COVID-19 emergency aid, 2021 | Yes | 104 | 22 (21.2) | 1.73 (1.01 - 2.95) | **0.044** |
| | No | 188 | 23 (12.2) | 1 | |
| **Occupational** | | | | | |
| Young apprentice | Yes | 6 | 0 | NA | NA |
| | No | 286 | 45 (15.7) | | |
| Number of works | Only one work/occupation | 236 | 35 (14.8) | 0.83 (0.44–1.57) | 0.569 |
| | More than one work/occupation | 56 | 10 (17.9) | 1 | |
| Work contract status | Formal work | 105 | 10 (9.5) | 1 | **0.045** |
| | Informal work | 187 | 35 (18.7) | 1.97 (1.01 - 3.81) | |
| Weekly workload | Up to 40 hours per week | 193 | 36 (18.7) | 1 | **0.026** |
| | More than 40 hours per week | 98 | 8 (8.2) | 0.44 (0.21–0.91) | |
| Work from home | Yes | 41 | 10 (24.4) | 1.74 (0.94 - 3.24) | **0.079** |
| | No | 250 | 35 (14.0) | 1 | |
| Being retired and working | Yes | 12 | 3 (25.0) | 1 | 0.326 |
| | No | 280 | 42 (15.0) | 0.60 (0.22–1.66) | |
| Work-related risk of exposure to SARS-CoV-2 [2] | Very high or high risk | 13 | 1 (7.7) | 0.43 (0.06–2.94) | 0.574 |
| | Medium risk | 184 | 27 (14.7) | 0.81 (0.47–1.41) | |
| | Low risk | 94 | 17 (18.1) | 1 | |
| Income changes during the pandemic | No change | 100 | 12 (12.0) | 0.77 (0.33–1.83) | 0.463 |
| | Decreased | 145 | 26 (17.9) | 1.15 (0.54–2.48) | |
| | Increased | 45 | 7 (15.6) | 1 | |
| Same work as before the pandemic | Yes | 208 | 34 (16.3) | 1 | 0.491 |
| | No | 84 | 11 (13.1) | 0.80 (0.43–1.51) | |

*(Continued)*

**Table 2.** (Continued)

| Characteristics | Categories | Number of participants | Number (%) with inadequate work ability | Crude PR (95% CI) [1] | P value [1] |
|---|---|---|---|---|---|
| **Health-related** | | | | | |
| Self-assessment of health | Excellent or very good | 79 | 2 (2.5) | 1 | **<0.001** |
| | Good or Moderate | 197 | 32 (16.2) | 6.42 (1.58 - 26.14) | |
| | Bad or very bad | 16 | 11 (68.8) | 27.16 (6.65-110.96) | |
| Body mass index range | Low and normal weight | 98 | 18 (18.4) | 1.31 (0.76–2.28) | 0.330 |
| | Overweight and obesity | 186 | 26 (14.0) | 1 | |
| Smoking | Yes | 20 | 1 (5.0) | 1 | 0.233 |
| | No | 272 | 44 (16.2) | 3.24 (0.47 −22.78) | |
| Hypertension | Yes | 74 | 19 (25.7) | 2.15 (1.27 - 3.66) | **0.005** |
| | No | 218 | 26 (11.9) | 1 | |
| Diabetes | Yes | 18 | 8 (44.4) | 3.29 (1.81 - 5.98) | **<0.001** |
| | No | 274 | 37 (13.5) | 1 | |
| Self-reported covid-19 illness | Yes | 104 | 20 (19.2) | 1.45 (0.85 - 2.47) | **0.178** |
| | No | 188 | 25 (13.3) | 1 | |

NOTE: NA, not applicable as it was not possible to estimate.

[1] Bivariate Poisson regression with robust variance was used to estimate prevalence ratios (PR), 95% confidence intervals (CI) and P values.

[2] The work-related risk of exposure to SARS-CoV-2 was defined in the methods section.

95% CI: 1.13–3.90; P = 0.02). Notably, the effect of sex on the occurrence of inadequate work ability was reduced after incorporating the occupational and, in particular, health-related variables (models B and C, Table 3), indicating that the effect of sex on work ability is at least partially mediated by these factors. The Akaike Information Criterion (AIC) for each model showed that model C, which included sociodemographic, occupational, and health-related variables, provided the best fit (Table 3).

## Discussion

Our study, conducted in a low-income urban community in Salvador, Brazil, two years after the onset of the COVID-19 pandemic, found that 57.9% of residents aged 16 and older were working, with about a quarter having changed their work during the pandemic. The proportion of working individuals in our study was similar to the occupation rate (including both paid and unpaid work, the latter in support of a family member's work) in the working-age population in Brazil and Salvador during the same study period (56.8% and 54.7%, respectively) [39].

However, the working participants differed from those who were not working in terms of sociodemographic and health-related characteristics. Women, individuals aged ≥45 years, those with fewer years of education, and those with a higher frequency of comorbidities were less likely to engage in paid work. These differences may reflect gender disparities, labor market discrimination, and cultural factors that overwhelm women with domestic duties and assign them caregiving roles in families. As a result, women often face challenges in balancing paid work with these essential but time-consuming responsibilities. In addition, these disparities highlight significant barriers for older individuals, those with less education, and those with chronic health conditions in securing and maintaining employment.

We also found that most (84.6%) of the study participants who were working had adequate working ability, with mean and median WAI scores similar to those reported in studies from several countries, such as Iran, Brazil, Italy, and Poland, which included workers from various sectors [40–43]. However, none of these studies were conducted in socially vulnerable populations. Therefore, the similarity between the WAI from our study and those from other settings may suggest that, despite the high level of social deprivation in our population, their work ability was likely not affected. Alternatively,

**Table 3. Multivariable hierarchical model analyses of factors associated with inadequate work ability among workers, Salvador, Brazil, 2022.**

| Characteristic | Model A. Sociode-mographic variables | Model B. Sociodemographic and occupational variables | Model C. Sociodemographic, occu-pational, and health-related variables |
|---|---|---|---|
| | **Adjusted PR (95% CI) and model AIC** | | |
| **AIC** | 257.502 | 251.788 | 229.395 |
| **Sociodemographic** | | | |
| Sex | | | |
| Male | 1 | 1 | 1 |
| Female | 1.89 (1.02 - 3.48) | 1.75 (0.95 - 3.23) | 1.27 (0.70 - 2.31) |
| Age | | | |
| <45 Years | 1 | 1 | 1 |
| ≥45 Years | 1.69 (0.99 - 2.89) | 1.61 (0.94 - 2.76) | 1.52 (0.90–2.58) |
| **Occupational characteristic** | | | |
| Weekly workload | | | |
| Up to 40 hours per week | | 1 | 1 |
| More than 40 hours per week | | 0.47 (0.28–0.96) | 0.53 (0.26–1.08) |
| **Health-related** | | | |
| Self-assessment of health | | | |
| Excellent or very good | | | 1 |
| Good or moderate | | | 5.91 (1.45 - 24.05) |
| Bad or very bad | | | 21.62 (5.14 - 90.91) |
| Diabetes | | | |
| No | | | 1 |
| Yes | | | 2.10 (1.13 - 3.90) |

the economic crisis induced by the COVID-19 pandemic may have acted as a filter, retaining individuals with better work ability in the workforce, while those with lower work ability may have been more vulnerable to job loss. Furthermore, the economic crisis may have introduced a positive bias in the assessment of work ability, as those who were able to maintain their jobs may have experienced greater job satisfaction, influencing their WAI scores. Hence, it is important to consider whether the high frequency of adequate work ability is, at least in part, due to a "healthy worker effect", where those who remain working tend to be healthier and better able to work. Future studies examining work ability should take the socio-economic context into account when interpreting the WAI.

In general, higher WAI scores are expected in younger people, while lower scores are typically observed in older individuals [19,22,34,35,44–46]. Although our study did not find a significant association between age and work ability, we observed a trend in the expected direction. After adjusting for sex, the prevalence ratio of inadequate work ability for those aged ≥45 years compared to those <45 years was 1.69, with a 95% CI ranging from 0.99 and 2.89. This suggests that such an association may exist in the population from which our sample was drawn. However, the sample size may not have been large enough to conclusively demonstrate this association, in contrast to another study conducted during the pandemic that did find such an association [44].

Regarding the role of sex and gender in work ability, women had an 89% higher prevalence of inadequate work ability compared to men. However, after adjusting for occupational and health-related characteristics, this effect was reduced to 27% and lost statistical significance. This suggests that the disparities in work ability between women and men are primar-ily explained by gender-related inequalities in occupational exposures and, particularly, health conditions, rather than by biological sex differences. Previous studies have similarly reported higher prevalence of inadequate work ability among women, attributing this to gendered factors, such as the unequal distribution of family responsibilities, greater exposure to

physical and mental stress associated with extended work hours, and the cumulative burden of multiple roles that women often fulfill [33,47–49].

Our hierarchical analysis further supports that workload and lower self-rated health mediate the association between gender and work ability. Moreover, in our sample, all domestic tasks and unpaid family caregiving were performed exclusively by women, underscoring the persistent asymmetry in the gender-based division of unpaid labor [50]. This pattern reflects broader structural gender inequalities in the labor market, including occupational segregation, limited access to leadership positions, wage disparities, and inadequate institutional support for balancing paid work with caregiving responsibilities. These systemic barriers not only restrict women's job opportunities but also negatively affect their health and long-term work capacity. Addressing such structural barriers is essential to promoting equity in employment and improving women's work ability outcomes. Despite social progress in recent decades, many women continue to be socially and culturally conditioned to assume caregiving and household roles [51], limiting their time and opportunities for paid work.

We also found that workers earning up to BRL 1,000 (USD 210) per month had a significantly higher prevalence of inadequate work ability (PR: 1.76). Although the PR remained relatively stable after adjusting for age and sex during the selection of variables from the first block (PR: 1.68), statistical significance was lost and the variable was not retained in the model. However, the existence of an association between income and work ability cannot be ruled out, as a bidirectional relationship between these factors is plausible. Prospective studies should explore this connection, as lower earnings may increase the risk of engaging in work activities that impair work ability, while inadequate work ability may, in turn, lead to lower earnings—potentially creating a vicious cycle. The non-significant but positive association of inadequate work ability with both being a recipient of Bolsa Família and the 2021 COVID-19 emergency aid further supports a relationship between income and work ability, as beneficiaries of these cash transfer programs typically have lower incomes.

In the bivariate analysis of occupational variables, informal workers – who do not contribute to the national social security program and lack legal rights to vacation, paid sick leave, and an end-of-year bonus – had a significantly higher prevalence of inadequate work ability than formal workers. However, this association did not hold in the multivariable analyses. The link between precarious working conditions and lower work ability had been observed before the pandemic [33]. In addition, a separate study that tracked work ability over 12 months during the pandemic, focusing on a sample predominantly composed of public servants with job stability and remote work arrangements, found that work ability remained largely stable. This finding suggests that more favorable working conditions may positively affect work ability [41]. Future studies should further explore the impact of employment insecurity, lack of rights, and limited protection on work ability.

The analysis of other occupational factors revealed an unexpected association: workers with a regimen of over 40 hours per week were less likely to experience inadequate work ability than those working 40 hours or fewer. This association, which persisted after adjusting for age and sex, contradicts previous studies that suggest excessive workload negatively impacts work capacity through effects like fatigue and stress [40,46]. A likely explanation for our finding is the presence of selection bias, specifically the "healthy worker effect", whereby individuals with adequate work ability are more likely to take on or be selected for longer work hours. In other words, work capacity may have influenced workload rather than the reverse. Further research is needed to clarify the complex relationships between work regimen characteristics, including contract type (formal vs. informal), and work ability across various occupational contexts.

Regarding the health-related variables, it is well established that chronic diseases can significantly impact work capacity [19,21,30]. Although hypertension was initially associated with inadequate work ability, this relationship did not remain significant in the multivariable model, likely due to high correlations with other health-related variables such as prior diabetes diagnosis and self-rated health status. Notably, having diabetes and self-rating health as less than excellent or very good were the factors most strongly associated with inadequate work ability. This aligns with findings from previous research, which show that individuals with poor self-rated health had a sixfold increase in the prevalence of limited work ability compared with those who perceived themselves to be in good health [21]. Given the high prevalence of hypertension and diabetes in our study population [24,52] and in other working groups [21,44,53], promoting a lifestyle that

includes healthy eating, physical activity, and the proper diagnosis and management of these conditions is essential to support work capacity.

The finding that self-assessed health status was strongly associated with inadequate work ability reinforces using self-perceived health as a key indicator of work capacity, as suggested by other studies [54]. Furthermore, it underscores the importance of considering both objective clinical aspects and subjective health perceptions when evaluating work capacity and developing strategies to enhance workers' well-being and productivity.

We found no association between self-reported COVID-19 and work ability. In our study, virtually all participants had been vaccinated against COVID-19, and none of the self-reported COVID-19 cases required hospitalization, suggesting they experienced mild illness. Previous studies have shown that COVID-19 severity influences the risk of developing post-COVID-19 symptoms [55], which, in turn, have been associated with substantial impacts on work-related activities, including increased absenteeism and reduced work ability [56]. Therefore, the mild nature of COVID-19 cases in our study may explain the absence of an association between self-reported COVID-19 and work ability. Additionally, the WAI has primarily been used to assess the effect of chronic illness on work ability [30,40,57], and the impact of acute mild illness on its score remains less clear. Similar to our findings, another Brazilian study following 1,211 workers found no differences in work ability between individuals infected and not infected with SARS-CoV-2, both at baseline (June to September 2020) and during follow-up (completed in October 2021) [41]. In contrast, a small study of 33 workers hospitalized for COVID-19 in Italy found a decline in work ability, with pre-COVID-19 WAI scores measured at hospital discharge dropping from good to medium levels one month later (mean WAI: 41.9 and 35.6, respectively) [58]. This decline was possibly driven by the severity of the disease in these patients.

This study has several limitations. First, due to its cross-sectional design, we were unable to establish causality for the observed associations. As mentioned earlier, reverse associations may have occurred, such as the relationship between workload and work ability. Second, the WAI instrument was originally designed to assess retirement eligibility related to aging and disability in the workforce [22,29], and it was not used for this purpose in our study. Moreover, since no specific cut-off points for the WAI have been validated for Brazilian workers [30], we relied on the cut-off points established by the instrument's developers. However, alternative cut-off points for younger populations have been proposed [59]. We also acknowledge that the WAI instrument is not comprehensive in capturing psychological, lifestyle, and contextual dimensions of work ability, as it does not account for factors such as stress, burnout, alcohol consumption, physical activity, job insecurity or social support, each of which is essential to a multidimensional assessment of work ability.

Third, we did not address relevant contextual factors related to work, such as working conditions, working environment, prior professional experience, or the above-mentioned lifestyle factors, all of which have been shown to influence work ability, both before and during the pandemic [43–46]. Additionally, data on COVID-19 history were collected via self-report, which may have led to misclassification, as some participants may have mistakenly identified other respiratory or febrile illnesses as COVID-19. Conversely, using a laboratory-confirmed diagnosis could have resulted in an underreporting of cases, given the limited availability of diagnostic tests, especially during the early months of the pandemic when testing was prioritized for severe cases. We also did not explore the relationship between post-COVID-19 symptoms or long COVID-19 and work ability, preventing a more comprehensive analysis of the effects of COVID-19 and work capacity.

Furthermore, the pandemic context may have influenced participants' perceptions of their work ability, as maintaining paid work during the socioeconomic crisis could have led to a more positive self-assessment of their overall capacity to work. Lastly, the relatively small sample size limited our power to detect weaker associations as significant and resulted in wide 95% confidence intervals for some prevalence ratios, indicating low precision in their measurement. Nevertheless, our study provides valuable exploratory insights into potential factors contributing to inadequate work ability within the context of urban vulnerability and a global health crisis.

## Conclusions

In summary, we found that nearly 15% of the workers from a low-income urban community in Brazil had inadequate work ability. This study was conducted amid the social and economic crisis caused by the COVID-19 pandemic, which may have influenced the profile of the working population and their capacity to work. The multidimensional hierarchical model used to explore associations between sociodemographic, occupational, and health-related characteristics with inadequate work ability revealed that these three dimensions can significantly impact work capacity. Notably, gender and health-related characteristics were identified as the most significant contributors.

These findings underscore the need for comprehensive policies that promote health, reduce social and occupational inequities, and ensure access to timely healthcare and effective management of chronic conditions to sustain work ability. Targeted interventions are particularly urgent for women, workers in precarious employment, and those living with chronic illness. Strengthening public health strategies—especially Primary Health Care with flexible hours and integrated occupational health services—is critical for the early identification and management of work-related risks and chronic diseases. Gender-responsive policies must also address the unequal burden of unpaid caregiving and fragmented work trajectories, while expanding access to social protection, supporting women's entrepreneurship, and fostering their participation in leadership and decision-making spaces. For informal and platform-based workers, it is essential to establish enforceable occupational health and safety standards and incentivize formalization through fiscal and regulatory measures. Finally, supporting the collective organization of informal and self-employed workers is key to ensuring their representation in the design of fairer and more inclusive labor policies.

## Supporting information

**S1 Database. Excel database containing the data analyzed for the manuscript "*Work ability during the COVID-19 pandemic: a cross-sectional study in a low-income urban setting in Brazil*".**
(XLS)

**S1 File. Codebook for the database analyzed for the manuscript *"Work ability during the COVID-19 pandemic: a cross-sectional study in a low-income urban setting in Brazil"*.**
(PDF)

## Acknowledgments

We thank all the residents of Alto da Pombas for their support for the researchers in conducting the study, especially those who generously agreed to participate. We also extend our thanks to the community health workers and other members of the local Family Health Program, students from the Universidade Federal da Bahia, and the technical staff from Fundação Oswaldo Cruz, who assisted the research team with field activities, data collection, and data management.

## Author contributions

**Conceptualization:** Ana Paula Cândido Oliveira, Daniela Alencar Vieira, Kionna Oliveira Bernardes Santos, Guilherme Sousa Ribeiro.

**Data curation:** Ana Paula Cândido Oliveira, Daniela Alencar Vieira, Guilherme Sousa Ribeiro.

**Formal analysis:** Ana Paula Cândido Oliveira, Guilherme Sousa Ribeiro.

**Investigation:** Ana Paula Cândido Oliveira, Daniela Alencar Vieira, Cristiane Wanderley Cardoso, Tereza Magalhães, Rosangela Oliveira Anjos, Eduardo José Farias Borges Reis, Kionna Oliveira Bernardes Santos, Guilherme Sousa Ribeiro.

**Methodology:** Ana Paula Cândido Oliveira, Tereza Magalhães, Kionna Oliveira Bernardes Santos, Guilherme Sousa Ribeiro.

**Resources:** Ana Paula Cândido Oliveira, Daniela Alencar Vieira, Cristiane Wanderley Cardoso, Eduardo José Farias Borges Reis, Kionna Oliveira Bernardes Santos, Guilherme Sousa Ribeiro.

**Software:** Ana Paula Cândido Oliveira.

**Supervision:** Guilherme Sousa Ribeiro.

**Visualization:** Ana Paula Cândido Oliveira, Guilherme Sousa Ribeiro.

**Writing – original draft:** Ana Paula Cândido Oliveira, Guilherme Sousa Ribeiro.

**Writing – review & editing:** Ana Paula Cândido Oliveira, Daniela Alencar Vieira, Cristiane Wanderley Cardoso, Tereza Magalhães, Rosangela Oliveira Anjos, Eduardo José Farias Borges Reis, Kionna Oliveira Bernardes Santos, Guilherme Sousa Ribeiro.

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
