## [Decision Letter · Decision Letter 0]

PONE-D-25-01676Work ability during the COVID-19 pandemic: a cross-sectional study in a low-income urban setting in BrazilPLOS ONE

Dear Dr. Ribeiro,

Thank you for submitting your manuscript to PLOS ONE. After careful consideration, we feel that it has merit but does not fully meet PLOS ONE’s publication criteria as it currently stands. Therefore, we invite you to submit a revised version of the manuscript that addresses the points raised during the review process.

We look forward to receiving your revised manuscript.

Kind regards,

Manuela Mendonça Figueirêdo Coelho, Ph.D

Academic Editor

PLOS ONE

Journal Requirements:

Reviewers' comments:

Reviewer's Responses to Questions

**Comments to the Author**

Reviewer #1: Yes

Reviewer #2: Yes

2. Has the statistical analysis been performed appropriately and rigorously? 

Reviewer #1: Yes

Reviewer #2: Yes

3. Have the authors made all data underlying the findings in their manuscript fully available?

Reviewer #1: Yes

Reviewer #2: Yes

4. Is the manuscript presented in an intelligible fashion and written in standard English?

Reviewer #1: Yes

Reviewer #2: Yes

5. Review Comments to the Author

Reviewer #1: Overall, the manuscript presents a well-structured and methodologically sound study. the topic is relevant, particularly given the socioeconomic disparities exacerbated by the COVID-19 pandemic. Using the Work Ability Index (WAI) and a hierarchical analytical model strengthens the study's validity. However, the following areas require improvement:

1. Methodology in terms of Clarity and Justification, Under the following Headings:

a. Sampling: the selection of 630 individuals from a total population of 5,478 is mentioned, but the exact randomization method should be detailed. Also, Clarify whether any weighting was used to ensure representativeness.

b. WAI: Explain why the cut-off point of 37 was chosen instead of alternative categorization approaches.

c. Statistical Analysis: Justify the use of Poisson regression over other models and explain the rationale for the backward elimination approach in selecting variables.

2. Result: Statistical Reporting and Interpretation under the following:

a. Prevalence Ratios: Maintain consistency in decimal places when reporting confidence intervals and include p-values alongside prevalence Ratio estimates for clarity.

b. Work Hours and Work Ability: The finding that longer work hours are associated with better work ability contradicts existing literature. Consider discussing the potential role of selection bias.

c. COVID-19 History: The absence of an association between prior COVID-19 and work ability should be explored in greater depth-particularly in the context of long COVID.

3. Discussion:

a. Discuss how the findings compare to research on work ability in other socioeconomic contexts.

b. The finding that gender differences in work ability are mediated by occupational and health factors aligns with existing research, but additional discussion on structural barriers for women in the labour market is needed.

c. The study has highlighted the importance of targeted intervention, but specific recommendations for policymakers and employers should be elaborated upon.

d. The cross-sectional design used in the study limits causal inference - this should be emphasized. Furthermore, the WAI's limitations in capturing psychological and contextual factors should be acknowledged.

Reviewer #2: The title of the manuscript was properly worded. The contents speaks to the objectives and data collection and analyses.

Abstract contained standard areas: brief introduction, problem, method, results and the contribution to knowledge.

Discussion: The author to concentrate on the impact of the study rather writing lengthy and many paragraphs on the limitation of study

he author to replace the following: Replace words like gender with sex, or any words speaking to the equity or diversity.

All my comments can be found on the body of the manuscript.

If all these minor concerns could be addressed, I recommend acceptance.

6. PLOS authors have the option to publish the peer review history of their article (what does this mean? ). If published, this will include your full peer review and any attached files.

**Do you want your identity to be public for this peer review?** For information about this choice, including consent withdrawal, please see our Privacy Policy .

Reviewer #1: **Yes: ** Lawan Adamu

Reviewer #2: No

---

## [Author Response · Author response to Decision Letter 1]

5 Jun 2025

RESPONSE TO THE EDITOR AND REVIEWERS

COMMENTS TO THE AUTHOR:

EDITOR COMMENT

Comment 1: Manuscript Formatting

Response:

We appreciate the reminder. The manuscript has been revised to fully comply with PLOS ONE’s style requirements, including file naming conventions. The formatting follows the journal’s guidelines.

Comment 2: Data Not Shown

We note that you have included the phrase “data not shown” in your manuscript. Unfortunately, this does not meet our data sharing requirements. PLOS does not permit references to inaccessible data. We require that authors provide all relevant data within the paper, Supporting Information files, or in an acceptable, public repository. Please add a citation to support this phrase or upload the data that corresponds with these findings to a stable repository (such as Figshare or Dryad) and provide and URLs, DOIs, or accession numbers that may be used to access these data. Or, if the data are not a core part of the research being presented in your study, we ask that you remove the phrase that refers to these data.

Response:

We thank the editor for this observation. We clarify that the results referred to by the phrase “data not shown” are, in fact, presented in the main text of the manuscript. These data were not included in Table 3 because the table reports only the variables retained in the final multivariable model. However, the relevant information is discussed in the Results section to ensure transparency and completeness. To align with PLOS data sharing policies, we have now removed the phrase “data not shown” and ensured that all referenced data are fully accessible in the text.

Comment 3: Reference list

Response:

We thank the editor for this observation. We revised and reformatted the entire reference list to comply with the journal’s style guidelines. We also carefully reviewed all references cited in the manuscript and confirmed that none of them have been retracted.

During this process, we removed the following references because they are academic theses/dissertations, and their findings are already supported by peer-reviewed publications included in the manuscript, such as Cadiz DM, Brady G, Rineer JR, Truxillo DM. A Review and Synthesis of the Work Ability Literature. Work, Aging and Retirement. 2019;5: 114–138. doi:10.1093/workar/way010.

Removed references as they reflected non-peer-reviewed publications:

• Milani D. Capacidade para o trabalho, sintomas osteomusculares e qualidade de vida entre operadores de máquinas agrícolas. Universidade Federal de São Carlos – UFSCar, 2011.

• Linhares JE. Avaliação da capacidade para o trabalho: análise frente ao envelhecimento funcional de servidores públicos em um município da região sul. Universidade Tecnológica Federal do Paraná – UTFPR, 2017.

• Oliveira Júnior P. Índice de capacidade para o trabalho (ICT): uma avaliação da capacidade laboral dos profissionais de enfermagem portadores de doenças crônicas não transmissíveis [dissertation]. Uberlândia: Universidade Federal de Uberlândia; 2018. doi:10.14393/ufu.di.2018.940

We also identified that the following references were duplicated, being listed twice. We removed the duplicated reference from the list.

• Burdorf A, Porru F, Rugulies R. The COVID-19 (Coronavirus) pandemic: consequences for occupational health. Scand J Work Environ Health. 2020;46: 229–230. doi:10.5271/sjweh.3893

• Hunter JR, Meiring RM, Cripps A, Suppiah HT, Vicendese D, Kingsley MI, et al. Relationships between Physical Activity, Work Ability, Absenteeism and Presenteeism in Australian and New Zealand Adults during COVID-19. Int J Environ Res Public Health. 2021;18: 12563. doi:10.3390/ijerph182312563

Lastly, we added the following references to support the discussion section as requested by Reviewer 1 in Comment 6.

• Wang R, Lin M, Yu S, Xue X, Hu X, Wang Z. Predictors of post-COVID-19 syndrome: a meta-analysis. J Infect Dev Ctries. 2025 Apr 29;19(4):490-497. doi: 10.3855/jidc.18574. PMID: 40305533.

• Ottiger M, Poppele I, Sperling N, Schlesinger T, Müller K. Work ability and return-to-work of patients with post-COVID-19: a systematic review and meta-analysis. BMC Public Health. 2024 Jul 7;24(1):1811. doi: 10.1186/s12889-024-19328-6. PMID: 38973011; PMCID: PMC11229229.

These adjustments have been made to ensure that the reference list is complete, accurate, and aligned with the journal’s requirements.

REVIEWER #1

General comment:

Overall, the manuscript presents a well-structured and methodologically sound study. The topic is relevant, particularly given the socioeconomic disparities exacerbated by the COVID-19 pandemic. Using the Work Ability Index (WAI) and a hierarchical analytical model strengthens the study's validity.

Response:

We thank the reviewer for the positive and thoughtful evaluation of our manuscript. We appreciate the recognition of the study’s structure, methodological rigor, and relevance in light of socioeconomic disparities intensified by the COVID-19 pandemic. We also value the constructive suggestions for improvement and have addressed each of them in detail below.

Comment 1: Sampling (Methodology)

The selection of 630 individuals from a total population of 5,478 is mentioned, but the exact randomization method should be detailed. Also, clarify whether any weighting was used to ensure representativeness.

Response:

Thank you for your comment. We have revised the manuscript to provide additional detail regarding the sampling procedure and to clarify the use of weighting, as below (lines 100-114).

“To obtain a representative sample of the neighborhood population, we randomly selected 630 individuals from the 5,478 residents aged 16 years or older registered in the community’s FHS, distributed across 11 microareas. Sampling was stratified by microarea and weighted according to the proportion of the population in each microarea. Random selection was performed at the household level using Excel (Office 365®), based on the FHS household registry. Considering an average household size of approximately three individuals, we randomly selected a number of households equivalent to about one-third of the weighted estimated number of required individuals per microarea. All eligible individuals aged 16 or older in the selected households were invited to participate.

This study was part of a broader research project investigating health and living conditions during the COVID-19 pandemic. The initial sample size of 630 participants was calculated to provide approximately 4% precision for estimating events with a 50% prevalence at a 95% confidence level. However, due to logistical challenges and restrictions imposed by the pandemic, the final sample comprised 504 participants from 273 households, including 292 workers residing in 205 households.”

Comment 2: WAI Cut-off Point (Methodology)

Explain why the cut-off point of 37 was chosen instead of alternative categorization approaches.

Response:

We thank the reviewer for raising this point. The threshold score of 37 was adopted based on the conceptual framework of the Work Ability Index (WAI), which classifies scores into four levels: low (7–27), moderate (28–36), good (37–43), and excellent (44–49) [21]. This classification is widely applied in occupational health research.

In this study, we recategorized the scores into two levels—inadequate work ability (<37, combining the low and moderate categories) and adequate work ability (≥37, combining the good and excellent categories)—to facilitate interpretation and to align with the WAI's recommendations for intervention. Specifically, scores below 37 indicate the need for targeted measures to restore or improve work ability, whereas scores of 37 or higher reflect conditions where the focus should be on supporting and maintaining work capacity.

This dichotomization has been used in previous studies [35–37] and facilitates clearer identification of workers at risk. We have clarified this rationale in the revised manuscript (lines 157-161), where the supporting references have been explicitly emphasized.

Comment 3: Statistical Analysis (Methodology)

Justify the use of Poisson regression over other models and explain the rationale for the backward elimination approach in selecting variables.

Response:

Thank you for your comment. We used Poisson regression models with robust variance because this method allows direct estimation of prevalence ratios, which are more appropriate and interpretable in cross-sectional studies with moderately or highly prevalent outcomes than odds ratios obtained from logistic regression models. Logistic regression could overestimate the strength of association under these conditions, making it a less suitable choice. We revised the manuscript to clarify this point (lines 188-190).

The backward elimination strategy was adopted to remove within-block variables that did not contribute meaningfully to explaining the outcome, thereby reducing model complexity and minimizing redundancy due to correlations among conceptually related variables within the same domain (e.g., socioeconomic or health conditions). We revised the manuscript to clarify this point (lines 195-199).

Comment 4: Prevalence Ratios (Results)

Maintain consistency in decimal places when reporting confidence intervals and include p-values alongside prevalence ratio estimates for clarity.

Response:

Thank you for your observation. We reviewed and standardized the number of decimal places used in reporting confidence intervals to ensure consistency. In addition, we added p-values alongside all prevalence ratio estimates in the paragraph describing the Poisson regression results. These revisions are highlighted in yellow in the manuscript for clarity.

Comment 5: Work Hours and Work Ability (Results)

The finding that longer work hours are associated with better work ability contradicts existing literature. Consider discussing the potential role of selection bias.

Response:

Thank you for your insightful comment. We agree that selection bias is a likely explanation for this finding and have already discussed it. To further emphasize this point, we revised the paragraph, which now reads (lines 393-395):

“The analysis of other occupational factors revealed an unexpected association: workers with a regimen of over 40 hours per week were less likely to experience inadequate work ability than those working 40 hours or fewer. This association, which persisted after adjusting for age and sex, contradicts previous studies that suggest excessive workload negatively impacts work capacity through effects like fatigue and stress [40,46]. A likely explanation for our finding is the presence of selection bias, specifically the “healthy worker effect”, whereby individuals with adequate work ability are more likely to take on or be selected for longer work hours. In other words, work capacity may have influenced workload rather than the reverse. Further research is needed to clarify the complex relationships between work regimen characteristics, including contract type (formal vs. informal), and work ability across various occupational contexts.”

Comment 6: COVID-19 History and Work Ability (Results)

The absence of an association between prior COVID-19 and work ability should be explored in greater depth—particularly in the context of long COVID.

Response:

Thank you for this valuable comment. The originally submitted manuscript discussed the lack of association between prior COVID-19 and work ability, contextualizing our findings within the existing literature. We have now further emphasized that this absence of association may be due to the predominantly mild COVID-19 cases in our study. Previous studies have shown that COVID-19 severity influences the risk of developing post-COVID-19 symptoms (Wang et al, 2025), which are, in turn, linked to substantial impacts on work-related activities, including increased absenteeism and reduced work ability (Ottiger et al, 2024). This clarification has been incorporated to the revised manuscript (lines 420-424). The following references have been added to support this context:

• Wang R, Lin M, Yu S, Xue X, Hu X, Wang Z. Predictors of post-COVID-19 syndrome: a meta-analysis. J Infect Dev Ctries. 2025 Apr 29;19(4):490-497. doi: 10.3855/jidc.18574. PMID: 40305533.

• Ottiger M, Poppele I, Sperling N, Schlesinger T, Müller K. Work ability and return-to-work of patients with post-COVID-19: a systematic review and meta-analysis. BMC Public Health. 2024 Jul 7;24(1):1811. doi: 10.1186/s12889-024-19328-6. PMID: 38973011; PMCID: PMC11229229.

Comment 7: Comparison with Other Contexts (Discussion)

Discuss how the findings compare to research on work ability in other socioeconomic contexts.

Response:

Thank you for your comment. We have already addressed this point in the third paragraph of the Discussion section, where we compare our findings with those from studies assessing work ability across different occupational sectors and in various countries (including Iran, Brazil, Italy, and Poland) (lines 320–3354. This discussion highlights that, despite similarities in WAI scores, none of the referenced studies were conducted in socially vulnerable populations like ours. The unexpectedly high work ability in our context allowed us to propose possible explanations, such as a selection bias, where the COVID-19 economic crisis may have filtered work opportunities toward those most able to work. These considerations help situate our findings within broader socioeconomic contexts.

Comment 8: Gender Differences (Discussion)

The finding that gender differences in work ability are mediated by occupational and health factors aligns with existing research, but additional discussion on structural barriers for women in the labour market is needed.

Response:

Thank you for your thoughtful comment. We agree that gender-related structural barriers in the labor market play a critical role in shaping gender differences in work ability. In response, we expanded the discussion in the revised manuscript to more explicitly address these barriers, including the unequal distribution of unpaid caregiving responsibilities, occupational segregation, wage disparities, and limited institutional support for balancing paid work and caregiving. These points help contextualize our findings within broader gender inequalities in the labor market (lines 344-365).

Comment 9: Policy and Employer Recommendations (Discussion)

The study has highlighted the importance of targeted intervention, but specific recommendations for policymakers and employers should be elaborated upon.

Response:

Thank you for your valuable suggestion. In response, we revised the conclusion to incorporate concrete recommendations for both policymakers and employers. Specifically, we added the need to strengthen public health services, particularly Primary Health Care, with flexible hours and integrated occupational health services, to enable early detection and management of work-related risks and chronic conditions.

We also emphasized the importance of gender-responsive policies that address the unequal burden of unpaid caregiving, improve access to social protection, and support women’s entrepreneurship and participation in leadership roles. For workers in informal or platform-based employment, we recommended establishing clear occupational health and safety standards, promoting formalization through fiscal and regulatory incentives, and supporting the collective organization of these workers to ensure their representation in

---

## [Editor Report · Decision Letter 1]

Work ability during the COVID-19 pandemic: a cross-sectional study in a low-income urban setting in Brazil

PONE-D-25-01676R1

Dear Dr. Ribeiro,

We’re pleased to inform you that your manuscript has been judged scientifically suitable for publication and will be formally accepted for publication once it meets all outstanding technical requirements.

Kind regards,

Manuela Mendonça Figueirêdo Coelho, Ph.D

Academic Editor

PLOS ONE
---

## [Editor Report · Acceptance letter]

PONE-D-25-01676R1

PLOS ONE

Dear Dr. Ribeiro,

I'm pleased to inform you that your manuscript has been deemed suitable for publication in PLOS ONE. Congratulations! Your manuscript is now being handed over to our production team.

Kind regards,

on behalf of

Dr. Manuela Mendonça Figueirêdo Coelho

Academic Editor

PLOS ONE